# Intergenerational Transmission of Child Feeding Practices

**DOI:** 10.3390/ijerph18158183

**Published:** 2021-08-02

**Authors:** Lilac Lev-Ari, Ada H. Zohar, Rachel Bachner-Melman, Auriane Totah Hanhart

**Affiliations:** 1Clinical Psychology Graduate Program, Ruppin Academic Center, Emek Hefer 4015000, Israel; ldlevari@gmail.com (L.L.-A.); adaz@ruppin.ac.il (A.H.Z.); auriane.hanhart@gmail.com (A.T.H.); 2Lior Zfaty Center for Suicide and Mental Pain Research, Emek Hefer 4015000, Israel; 3School of Social Work, Hebrew University of Jerusalem, Emek Hefer 4015000, Israel

**Keywords:** retrospective child feeding practices, current child feeding practices, eating disorder symptoms, Child Feeding Questionnaire, BMI

## Abstract

This study assessed the relationships between parents’ retrospective recollections of their mothers’ child feeding practices (CFP), current disordered eating (DE) and current CFP (how they now feed their children). 174 Israeli parents (136 mothers, 38 fathers; 40.1 ± 6.9 years of age) of children between the ages of 2 and 18, living at home, completed questionnaires online assessing demographics, retrospective recollections of the CFP that their mothers used when they were children, current CFP and current DE. Specific aspects of retrospectively recalled maternal CFP were significantly associated with the same aspects of current CFP. Current DE mediated the association between retrospectively recalled maternal CFP and current CFP and moderated the association between current concern about child’s weight and pressure for child to eat. Results highlight that the way adults pass on their feeding practices to their children is strongly influenced by their childhood recollections of their mothers’ concern about their weight, pressure for them to eat or restriction of their food intake. People often strive to behave differently from their parents, especially in the realm of food and eating. However, our findings suggest that parental CFP can become entrenched and can be passed on to our children.

## 1. Introduction

Child feeding practices (CFP) are behaviors adopted by parents to feed their children [1]. Parents use strategies to influence their children’s eating behaviors either directly or via modelling [2], some overt and some covert [3]. Strategies include attempting to control children’s food intake, pressuring them to eat, monitoring their intake and restricting their intake of fatty foods [4]. Research has shown that parents of underweight children tend to use overt strategies such as pressuring them to eat [5], whereas parents who perceive their children to be overweight tend to restrict their caloric intake in an attempt to control their children’s weight [1]. However, these strategies usually backfire, since they tend to foster fussy eating [6,7,8] and fail to change weight status [9], frustrating children and parents alike [10,11]. The causal direction remains unclear. CFP might modify eating behaviors and subsequently weight status of children. Alternatively, they may develop as a consequence of and in reaction to the child’s weight status.

Birch and Fisher [1] found that parents of high-weight children often restrict their children’s caloric intake. Powers et al. [12], however, found no link between parent feeding strategies and child weight status in socioeconomically and ethnically diverse cultures [12]. A recent systematic review and meta-analysis of controlling CFP and child weight found a small yet significant association between restrictive CFP and child weight and a significant association between pressure to eat and low child weight status. Parental perception of and concern about their children’s weight [13,14], as well as preoccupation with their own weight [15], may therefore be more pertinent to restrictive CFP than children’s actual weight status. Mothers’ mental health symptoms have also been shown to be associated with more controlling CFP. Webber et al. [16] assessed 7–9-year-olds and found that maternal perceptions of their child’s weight mediated the association between controlling CFP and child weight. As hypothesized, mothers of higher-weight children tended to restrict their children’s food intake, while mothers of lower-weight children tended to pressure them to eat. Moreover, maternal concern about their children’s weight fully mediated the positive association between child weight and maternal restrictive CFP. The main reason for maternal restrictive CFP therefore seemed to be mothers’ concern about their child’s weight.

Many studies have examined connections between CFP and child weight and between CFP and parents’ concern about their child’s weight. Yet, little research has addressed the long-term influence of CFP experienced by children on their weight and CFP as adults. Parental pressure for children to eat has been found to be associated with lower body mass index (BMI), and parental restriction of their children’s food intake with higher BMI when those children become adolescents [17]. College students reported that their recollections of parental CFP had influenced their current eating habits, such as regular eating, finishing all the food on their plates, and eating dessert [18]. Galloway et al. [19] found a positive association between retrospectively reported parental restrictive CFP and current BMI in college students. The stronger the students’ recollections of their parents restricting their food intake during childhood, the higher was their current BMI.

Two studies assessed adults’ retrospective recollections of CFP. One found no significant associations between college students’ retrospective CFP and current adult BMI [19]. The other found a significant difference between adult men and women in a community sample [9]. For men, a positive association was found between recollections of maternal concern for child’s weight and current BMI. For women, recollections of maternal concern for child’s weight, and restriction and monitoring of their food intake were all positively and significantly correlated with adult BMI. The authors concluded that even if retrospective recollections of CFP are biased, they appear to have long-lasting implications for adults’ eating practices and BMI over the lifespan [9].

A systematic review from 2014 [20] that identified seventeen studies examining parents’ current disordered eating (DE) in relation to their CFP towards their children concluded that there is a significant association between the two. Mothers with bulimic tendencies were found to restrict their daughters’, but not their sons’, food intake [21] and to pressure their children to eat [22]. Mothers and fathers dissatisfied with their bodies pressured their children to eat more than other parents [22]. Sadeh-Sharvit et al. [23] interviewed 29 mothers who had developed an eating disorder before they gave birth about CFP with their toddlers. These mothers reported being preoccupied with what their children ate, facing many dilemmas when feeding them, and experiencing extreme apprehension about their daughters’ shape and weight.

Two studies used community samples to examine the connection between parents’ retrospective recollections of their mothers’ CFP when they were children and current adult BMI and DE [9,24]. Positive correlations were observed between adult men and women’s current BMI and their recollections of maternal concern about their weight as children. Women’s current BMI was positively associated with their recollections of their mothers’ restriction and monitoring of their food intake when they were children. Men and women’s recollections of their mothers’ restriction of their food intake and concern for their weight as children was positively associated with their current DE, drive for thinness and body dissatisfaction as adults [9,24]. These findings highlight the need to further understand the connection between adults’ recollections of CFP during childhood and current DE.

To our knowledge, no studies to date have explored possible relationships between adults’ recollections of their mothers’ CFP during childhood, current DE and CFP towards their children. This was the aim of the present study. In our hypotheses listed below, we took into account that BMI has been shown to correlate strongly with both CFP and EDS:There will be a positive and significant association between retrospectively recalled maternal CFP and current CFP, after controlling for BMI.There will be positive and significant associations between parents’ current DE, current CFP and retrospectively recalled maternal CFP, after controlling for BMI.Parents’ current DE and BMI will mediate the association between their current CFP and their retrospectively recalled maternal CFP.Parents’ current DE will moderate the association between their current concern about their child’s weight and both pressure for him/her to eat and restriction of his/her caloric intake.

## 2. Method

### 2.1. Participants

The sample comprised 174 Israeli parents (136 mothers, 38 fathers) aged 26–54 years (mean = 40.05; SD = 6.85), who had at least one child between the age of 2 and 18 living with them at home. Only one parent per family (mother or father) was eligible to participate, so that no two participants reported about the same child. Children’s ages ranged between 2–18 (mean = 9.48; SD = 5.58). Participants’ years of education ranged between 10 and 26 years (mean = 16.20; SD = 2.77), indicating that they were a relatively educated sample from a relatively high socioeconomic background. BMI ranged between 15.76 and 45.44 (mean = 24.88; SD = 5.12) for women and between 21.45 and 44.98 (mean = 25.44; SD = 4.01) for men. Seven (4.1%) were low weight (BMI < 18.5), 144 (84.7%) were average weight (18.5 < BMI < 24.9) and 19 (11.2%) were high weight (BMI > 30).

### 2.2. Measures

*Child Feeding Practices* were assessed by the Child Feeding Questionnaire (CFQ) [25], a self-report measure that assesses parental beliefs, attitudes and behaviors connected to child feeding. Responses to its 31 items are recorded on a 5-point scale and form seven sub-scales; (1) perceived parental responsibility for child’s weight (e.g., “how often are you responsible for your child’s food portion during meal time?”), (2) perceived parental weight during parent’s childhood (e.g., “how would you estimate your weight during your adolescence?”), (3) perceived child’s weight (e.g., “how would you estimate your child’s weight in his/her early childhood {age 3–5}?”), (4) concern about child’s weight (e.g., how often are you preoccupied with the possibility that your child will be fat?”), (5) parental restriction of food intake (e.g., “I have to be sure that my child does not eat too many sweets such as candies, ice-cream, cake or pastries”), (6) parental pressure to eat (e.g., “my child should always eat all of the food on her/his plate”), and (7) parental monitoring of high-fat food consumption (e.g., “how much do you keep track of the high-fat food that your child eats?”). The questionnaire has been translated into Hebrew and all subscales had high internal consistency, with Cronbach’s alpha ranging between 0.73 and 0.91 [9]. In this study, the Cronbach’s alpha ranged between 0.67 and 0.91.

*Retrospective Child Feeding Practices* were assessed by the Retrospective Child Feeding Practices (RCFP) scale that assesses adults’ retrospective perception of their mothers’ CFP during their childhood [24]. The concern subscale consists of three items that assess mothers’ concern with their child’s weight. The original item “How concerned are you about your child eating too much when you are not around her?” was revised, for example, to “How concerned was your mother about your eating too much when she wasn’t around you?” The restriction subscale consists of eight items that assess the extent to which mothers used food as a reward and felt they had to watch or restrict their children’s caloric intake. The original item “I have to be sure that my child does not eat too many sweets” was revised, for example, to “my mother had to be sure that I didn’t eat too many sweets.” The pressure-to-eat subscale consists of four items that assess the extent to which mothers encouraged their children to eat more than they chose to eat. The original item: “My child should always eat all of the food on her plate” was revised, for example, to “I always had to eat all the food on my plate”. The monitoring subscale has three items that assess the extent to which mothers oversaw their child’s eating. The original item “How much do you keep track of the high-fat foods that your child eats?” was revised, for example, to “How much did your mother keep track of the high-fat foods that you ate?”. Cronbach’s alpha for the revised Hebrew subscales were 0.81 for maternal concern; 0.81 for maternal restriction; 0.73 for maternal pressure to eat; and 0.91 for maternal monitoring. The RCFQ has been translated to other languages and shown good construct validity [25,26].

*Disordered attitudes toward food and eating* were measured using the Eating Attitudes Test (EAT-26) [27]. This self-report questionnaire contains 26 questions divided into three subscales: Dieting (e.g., “I feel extremely guilty after eating”), Bulimia and Food Preoccupation (e.g., “Have gone on eating binges where I felt I may not be able to stop”) and Oral Control (e.g., “I avoid eating when I am hungry”). Responses are selected using a 6-point scale, ranging from 1 (never) to 6 (always), with higher scores reflecting more disordered eating attitudes. The questionnaire was translated into Hebrew by Apter and Yanko [28]. Cronbach’s in this study was high; 0.90 for Dieting, 0.75 for Bulimia and Food Preoccupation and 0.56 for Oral Control. 

### 2.3. Procedure

The study was approved by the Institutional Review Board. No funding was received for the study. Participants were recruited via social media, mostly via Facebook pages relevant to parenting and nutrition, intended for young, normative adults in a representative selection of geographical areas in Israel. Questionnaires were completed online using Qualtrics software (www.qualtrics.com). Prior to accessing the questionnaires, participants provided informed consent after reading an explanation about the purpose of the research, use of the data, and their right to withdraw from the study at any point.

### 2.4. Data Analysis

Descriptive statistics were used to quantify frequencies and Pearson correlations were used to quantify the associations between variables. The Structural equation model (SEM) was employed to assess the mediating effect of DE on the association between retrospectively reported and current CFP and to ascertain the mediating effects of DE and current CFP on the association between retrospectively reported maternal CFP. Statistical analyses were conducted using SPSS 23.0 and AMOS 23.0.

## 3. Results

**Hypothesis** **1.***There will be a positive and significant association between retrospectively recalled maternal CFP and current CFP, after controlling for BMI*. 

Table 1 shows the correlations between retrospectively recalled maternal CFP and current CFP, after controlling for BMI. As can be seen, there were significant, positive correlations between retrospectively recalled and current CFP that are specifically matched. Parents who remembered their mothers as being more concerned about their weight tended to be more concerned about their own children’s weight, and the same held for restriction of food intake, pressure to eat and monitoring of high-fat food consumption. Smaller but significant and positive correlations were also found between retrospectively recalled restriction of food intake and both current concern for child’s weight and monitoring of child’s food intake, and between retrospectively recalled pressure to eat and both current restriction and monitoring of food intake.

**Hypothesis** **2.***There will be positive and significant associations between parents’ current DE, current CFP and retrospectively recalled maternal CFP, after controlling for BMI*.

As can be seen in Table 2, positive and significant correlations were observed between parents’ current DE and current concern about child’s weight and restriction and monitoring of food intake. A significant, negative correlation was observed between parents’ current DE and pressure to eat. The more disordered the parents’ eating, the more concerned they were about their child’s weight, the more they restricted and monitored their children’s food intake and the less they pressured them to eat. There was also a significant, positive correlation between parents’ current DE and retrospectively recalled maternal concern for child’s weight and restriction and monitoring of food intake. The more parents recalled their mothers’ being concerned about their weight and restricting and monitoring their food intake as children, the more disordered were their current eating attitudes. Parents’ BMI was positively correlated with retrospective concern (r = 0.37, *p* < 0.001) and with retrospective restriction (r = 0.23, *p* < 0.001), meaning the higher the parents current BMI the more they remembered their mother being concerned with their weight as children and restricting their food intake. Parents’ BMI was also negatively correlated with current pressuring their children to eat (r = −0.30, *p* < 0.001), meaning the higher the parents current BMI the less likely they were to pressure their children to eat.

**Hypothesis** **3.***Parents’ current DE and BMI will mediate the association between their current CFP and their retrospectively recalled maternal CFP*.

An SEM was designed, following the recommendation of Hayes [29]. DE and BMI were assessed as mediating variables between retrospectively recalled maternal CFP and current CFP. As a combined rule for the acceptance of our model, we chose the following acknowledged values: normed fit index (NFI) > 0.90 [30] and root mean square error of approximation (RMSEA) < 0.08 [31] (see Figure 1). The Chi Square goodness-of-fit index presented an excellent fit for the data, $\chi_{\left(21\right)}^{2}\;$ = 39.26, *p* = 0.01); NFI = 0.92; CFI = 0.96; RMSEA = 0.07; standardized root means square residual (RMR) = 0.06.

Retrospectively recalled maternal concern about one’s weight as a child positively predicted current DE and negatively predicted current restriction and monitoring of children’s food intake. Retrospectively recalled maternal monitoring of one’s food intake as a child positively predicted current DE and monitoring of child’s food intake. Retrospectively recalled parental pressure to eat negatively predicted current BMI and current concern for child’s weight and positively predicted current pressure for one’s child to eat. Retrospectively recalled parental restriction of food intake as a child positively predicted current BMI and current restriction of child’s food intake. There was a significant, positive association between current BMI and DE. Current BMI negatively predicted current pressure for child to eat and monitoring of his/her high-fat food consumption. DE positively predicted only current concern about child’s weight. 

**Hypothesis** **4.***Parents’ current DE will moderate the association between their current concern about their child’s weight and both pressure for him/her to eat and restriction of his/her caloric intake*.

To examine interaction effects between parental DE and concern for child’s weight on pressure to eat, we used Hayes’s [29] process moderation analysis. DE did not significantly predict pressure for child to eat (β = −0.17, *p* = 0.13), but concern for child’s weight did (β = −0.21, *p* < 0.01), explaining 7.52% of the variance, *F*_(3,170)_ = 4.61, *p* < 0.01. The interaction between parental DE and concern for child’s weight was statistically significant (β = 0.20, *p* < 0.05), and the addition of this interaction to the regression model contributed a further 2.1% of explained variance, *F*_(1,170)_ = 3.78, *p* < 0.05. Figure 2 presents the interaction findings.

As can be seen in Figure 2, pressure placed on their child to eat by parents with *high* DE was *lower* than pressure placed on their child to eat by parents with *low* DE, regardless of parental concern for child’s weight. However, the pressure placed on their child to eat by parents with *low* DE depended on their concern for their child’s weight. Parents with *low* DE and *low* concern for their child’s weight placed *high* pressure on their child to eat. Parents with *low* DE and *high* concern for their child’s weight, placed *low* pressure on their child to eat. 

To examine interaction effects between parental DE and concern for child’s weight on restriction of caloric intake, we used Hayes’s [29] process moderation analysis. While concern for child’s weight positively predicted restriction of caloric intake (β = 0.27, *p* < 0.001), DE did not, and the moderation analysis was not statistically significant. Parents who were highly concerned about their child’s weight restricted their child’s caloric intake, regardless of their own DE.

## 4. Discussion

This study investigated adults’ retrospective memories of their mothers’ CFP when they were children in relation to their self-reported current DE, BMI and CFP towards their children. When there was more than one child under 18 living at home, parents were asked to report on the child they were most concerned about with issues of food and eating. Roberts et al. [32] showed that retrospective reports of CFP have been found to be reliable and to have long-term effects that extend well into adult life. Moreover, mothers and their adult daughters tend to have similar recollections of mothers’ CFP when their daughters were children.

Perhaps the most significant finding of this study is that specific feeding practices that participants retrospectively reported their mothers used when they were children tended to be those that they themselves currently used with their own children. Adults whose mothers expressed concern about their weight when they were children currently felt concern about their own children’s weight, those whose caloric intake was restricted by their mothers when they were children currently tended to restrict their children’s caloric intake, those whose mothers used to pressure them to eat now tended to pressure their children to eat and those whose mothers used to monitor their food intake carefully tended to do the same with their children.

There is evidence that adults’ (both men’s and women’s) retrospective memories of their mothers’ concern about their weight as children and attempts to restrict their food intake are associated with higher BMI, more DE and more body dissatisfaction later in life [9]. Mothers’ concern about their children’s weight and restriction of their food intake may develop as a reaction to their children’s weight or appetitive tendencies [33], out of a well-meaning desire to help them be healthy and conform to beauty ideals. This is not, however, the whole story, since concern for children’s weight and restrictive feeding practices have been shown to be associated with mothers’ BMI, body dissatisfaction and stigma against high weight [34], perpetuating or amplifying intergenerational transmission. A short longitudinal study by Webb and Haycraft [35] showed that maternal body dissatisfaction fostered restrictive CFP. Since children generally experience maternal concern about weight and restriction of their food intake as aversive, and these practices seem unsuccessful in changing BMI, it is unfortunate that mothers tend to transmit them to their children.

We found additional, though indirect, support for this transmission process. Participants with high DE tended not to pressure their children to eat, whether or not they were concerned about their weight. However, the picture was more complicated for parents with low DE. Parents with *low* DE and *low* concern for their children’s weight, tended to pressure their children to eat, whereas parents with *low* DE and *high* concern for their children’s weight, tended *not* to pressure their children to eat.

This pattern was not observed for parental restriction of their children’s caloric intake. Parents with high concern about their children’s weight (whether or not the concern was justified) tended to restrict their children’s caloric intake regardless of their own weight and eating attitudes. This finding supports studies that found that parental feeding practices were predicted by parental concern for children’s weight but not children’s actual BMI [16,36].

Galloway et al. [11] found that parental pressure to eat is usually experienced by children as aversive. Furthermore, they found that pressure to eat paradoxically *reduces* children’s food intake. It therefore seems particularly unfortunate that parents who recall being pressured to eat as children tend to pressure their own children to eat. Although pressuring a child to eat may be a well-intentioned parental response to child undereating and their fussy eating, longitudinal studies show that this pressure also predicts future food fussiness as children develop, amplifying rather than dampening the fussy eating [37]. Furthermore, pressuring children to eat is associated with picky eating in adulthood [38]. It therefore seems clear that this specific parental feeding strategy generally backfires.

Participants’ current level of DE was positively associated, beyond their BMI, with their current concern for their children’s weight, restriction of their children’s food intake and monitoring of their children’s consumption of high calorie foods. Their DE levels were also negatively associated with the pressure they placed on their child to eat. This pattern of correlations was similar for their retrospective reports of their own mothers’ feeding practices, so that retrospectively reported maternal CFP were also related to DE. Although data were cross-sectional, ruling out conclusions about causality, this pattern of correlations is compatible with a cycle of intergenerational transmission. Maternal CFP (albeit reported retrospectively) seem to have long-term implications for children’s eating attitudes and their eventual CFP with their own children. Retrospectively reported CFP and current CFP with one’s children are also associated with adult body dissatisfaction [9], possibly supporting a mechanism of intergenerational transmission of feeding practices and DE.

All these associations are evident in the SEM model we developed, which showed that retrospectively recalled feeding practices used by one’s mother are associated with current feeding practices directly, as well as indirectly via the mediation of current BMI and DE. Thus, intergenerational nurturing goes awry. Children told to finish their soup [11] might, as adults, hear themselves urging their own children to do just that. Children monitored and restricted because of their mother’s concern for their weight might find themselves as parents worrying about their children’s weight and restricting and monitoring their food intake, even though these approaches were not helpful for them. Moreover, these well-meant parental efforts have been found to be associated with less intuitive eating and more effortful control, which result, in turn, in more DE [32].

### Limitations

This study has several limitations. First, all data were self-reported, and therefore subjective. Direct observation in real time may have had more field validity. Second, this was a cross-sectional study that strove to add to our understanding of inter-generational transmission processes, which are best studied longitudinally. Third, the age of the children was not considered, and parental CFP may evolve as children develop. Fourth, children’s BMI was not measured or reported because the focus of our study was on parental perception of their children’s weight, which has direct implications for parental feeding practices, the focus of this study. Fifth, retrospective childhood recollections of child feeding practices used by the participants’ fathers was not reported, and these may have differed from child feeding practices used by their mothers. Sixth, the study was conducted with a well-educated Israeli sample of parents, and it is therefore unclear whether or not results can be generalized to parents and children of different nationalities and backgrounds. Finally, retrospective studies are obviously hindered by hindsight. We do not actually know that the CFP reported were those in fact adopted by participants’ mothers. Yet, once again, we were interested in parents’ *recollections* of how their mothers fed them, rather than in an objective reality. These very limitations are therefore also strengths in the sense that subjective beliefs were the focus of our interest. Future studies should take into consideration children’s ages, developmental changes in CFP and children’s actual weights. 

## 5. Conclusions

This study shows the high association between the way people remember their mother’s concern for their weight during childhood, pressure to eat and restriction of food intake and the transmission of these feeding practices to the next generation. Parents often strive to offer their children a different and better upbringing from the one they received, yet the CFP they recall receiving from their parents are those that they most end up implementing with their own children. High-weight parents often remember their own parents’ negative remarks about their body and weight as children and strive not to repeat this with their children. However, our findings show how integrated CFP tend to become in our thinking and behavior. It is to be hoped that clinicians who treat parents with DE will be able to use this knowledge to help these parents to liberate themselves from the perpetuation and intergenerational transmission of negative beliefs, approaches and behaviors about their children’s eating and weight, thereby improving their children’s chances of developing self-acceptance, a positive body image and healthy eating habits.

## Figures and Tables

**Figure 1 ijerph-18-08183-f001:**
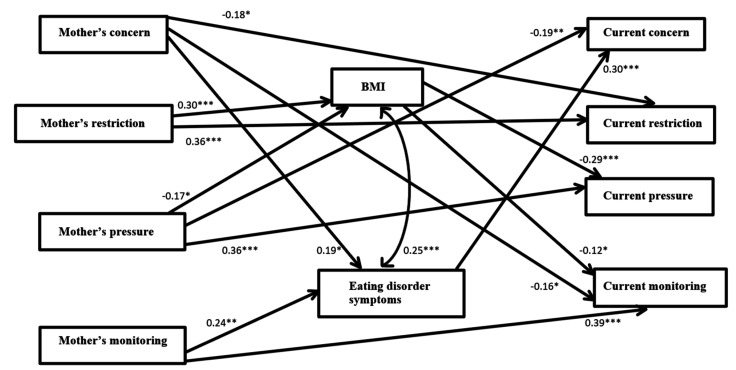
Path analysis: disordered eating and BMI as mediators between retrospectively recalled maternal CFQ and current CFQ. Note: Mother’s CFP relate to what the participant reported concerning their mother’s CFP when they were children. * *p* < 0.05; ** *p* < 0.01; *** *p* < 0.001; current CFP relates to the way participants feed their own children. Concern = concern for child’s weight; Restriction = restriction of food intake; Pressure = pressure to eat; Monitoring = monitoring child’s food intake.

**Figure 2 ijerph-18-08183-f002:**
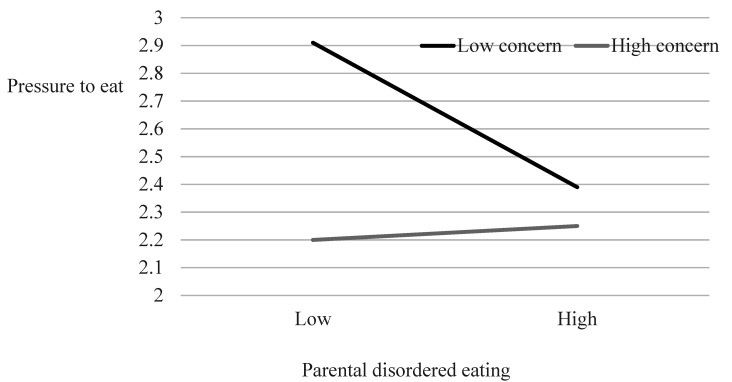
Interaction between parental disordered eating and concern for child’s weight on pressure for child to eat.

**Table 1 ijerph-18-08183-t001:** Pearson correlations between retrospectively recalled maternal CFP and current CFP, after controlling for BMI; subscale means and SDs (N = 174).

Current Retrospective	Concern	Restriction	Pressure	Monitoring	Mean (SD)
Concern	0.19 **	0.09	−0.08	0.09	1.93 (1.12)
Restriction	0.13 *	0.29 ***	0.05	0.17 *	2.06 (0.77)
Pressure	−0.11	0.22 **	0.37 ***	0.13 *	2.70 (1.06)
Monitoring	0.12	0.10	−0.07	0.28 ***	2.32 (1.13)
Mean (SD)	2.00 (1.00)	2.58 (0.73)	2.49 (0.99)	3.52 (0.89)	

Note: * *p* < 0.05; ** *p* < 0.01; *** *p* < 0.001 one tailed. Current = current CFP; Retrospective = retrospective CFP; Concern = concern about child’s weight; Restriction = restriction of food intake; Pressure = pressure to eat; Monitoring = monitoring of high-fat food consumption.

**Table 2 ijerph-18-08183-t002:** Pearson correlations between parent’s current disordered eating (EAT-26), current CFP and retrospectively recalled maternal CFP, after controlling for BMI; subscale means and SDs (N = 174).

	EAT-26
Current (CFQ)	
Concern	0.38 ***
Restriction	0.16 *
Pressure	−0.17 *
Monitoring	0.16 *
Retrospective (RCFQ)	
Concern	0.37 ***
Restriction	0.24 ***
Pressure	0.07
Monitoring	0.33 ***

Note: * *p* < 0.05; *** *p* < 0.001 one tailed. CFP = child feeding practices; Concern = concern about child’s weight; Restriction = restriction of food intake; Pressure =pressure to eat; Monitoring = monitoring of high-fat food consumption; EAT-26 = Eating Attitudes Test-26.

## Data Availability

Data are available from the authors on request.

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
