# Peer review of "Intergenerational Transmission of Child Feeding Practices"

_ijerph, 2021, doi:10.3390/ijerph18158183_

Round 1

Reviewer 1 Report

The authors investigated the intergenerational transmission of child feeding practices. I enjoyed reading and reviewing the short manuscript. However, I found the major flow lies in the Title which must be modified. Please see my comments below

L(ine) 1- Title? Authors are suggested to change/modify the title of the work. The first few words are pretty confusing and may not attract future readers. What does feeding indicate? Breastfeeding? Maternal feeding? Clarify and confirm. In fact, all participants were from only one country which indicates the study is more like a case study rather than generalizing. Although the whole theme of the research is super common and therefore extremely expected. To impress and to make it to the point- you may consider the following title.

Intergenerational transmission of child feeding practices: a case study on Israeli parents

L  10-11- Remove how…..children

L11- 1 decimal points, always.

L11- How about other parents? Men? Age? What is the ethnicity/ origin of the parents participated? Feeding does not indicate only breastfeeding. Are you indicating – Maternal parenting/ eating ONLY?

L 22-23- Was it not expected/ anticipated? This is common sense. Suggest to start the sentence as—However, as expected, our findings reassure that parental CFP entrenches in our parental life and can therefore be passed on to our next generation.

L29- Remove the first sentence of the Intro.

L 111-116- Again, how about male parents? Are you indicating breastfeeding? Make it CLEAR, right from the beginning (title).

L231- Fig 1. How did you prepare this? Is it your own? Or you followed the technique/ method from others and then inserted/applied your data/information to see what it says? If it is from others, please refer and cite them.

L259-260- Fig 2. Add both axes to the Fig. Revise the name of X-axis. Parental disordered eating itself should be enough to understand that it would be high as the value of X-axis increases.

L273- Fig 3. Same comments as given for Fig 2.

L345- what does maternal stand for? Why? Clarify.

L 369- Remove the first two words of Conclu

L376- Remove/ replace Sadly. This word is inappropriate for a study like yours.

*** Suggest to add/ provide all data as supplementary.

Author Response

Review 1

The authors investigated the intergenerational transmission of child feeding practices. I enjoyed reading and reviewing the short manuscript. However, I found the major flow lies in the Title which must be modified. Please see my comments below

  1. L(ine) 1- Title? Authors are suggested to change/modify the title of the work. The first few words are pretty confusing and may not attract future readers. What does feeding indicate? Breastfeeding? Maternal feeding? Clarify and confirm. In fact, all participants were from only one country which indicates the study is more like a case study rather than generalizing. Although the whole theme of the research is super common and therefore extremely expected. To impress and to make it to the point- you may consider the following title.

Intergenerational transmission of child feeding practices: a case study on Israeli parents

The first sentence has been deleted. What is now the first sentence defines the term “child feeding practices”, which is a well-known concept in research on nutrition, eating disorders and obesity, and adds a reference for clarity. Regarding the title, we feel that referring to our study as a “case study” would be misleading, since the reader may associate this term with single case descriptive studies, whereas we used a sample of 174 participants and conducted statistical analyses. After giving an alternative title some thought, we agreed among ourselves that we like the current title. If the reviewer (and Editor) feel it is important to stress that all participants were from Israel, the title could be “I will not be my mother: Intergenerational transmission of child feeding practices in Israel”.

In line 25 of the current manuscript, we have changed “their parents’ child feeding practices” to “their mothers’ child feeding practices”. Apologies for having made this error that we believe may have led you to misunderstand those recollections reported are only of mothers’ child feeding practices, not of fathers’ child feeding practices. In line 29 we have also spelled this out by changing the expression “maternal child feeding practices” to “the child feeding practices that their mothers used when they were children”.

  1. L  10-11- Remove how…..children

We added this simple explanation both to clarify the notion of child feeding practices in line with your comment no. 1 above, and to clarify the term “current child feeding practices” as opposed to retrospective recollections of maternal child feeding practices. We believe that it is a helpful addition, however if you and the Editor are in favor of deleting it, please do that.

  1. L11- 1 decimal points, always.

This has been corrected.

  1. L11- How about other parents? Men? Age? What is the ethnicity/ origin of the parents participated? Feeding does not indicate only breastfeeding. Are you indicating – Maternal parenting/ eating ONLY?

We have now stipulated that participants were 136 mothers and 39 fathers, and their mean age and standard deviation is presented. We have added the word “Israeli” to present their origin. “Retrospectively recalled maternal child feeding practices” refers to the participants’ memories of their mothers’ child feeding practices. We have changed this to “retrospective recollections of maternal child feeding practices when they were children” to spell this out. If you think it is not clear enough “not breastfeeding” can be added in parentheses. However, since people can generally not recall being breastfed because of their young age, we believe that the addition is superfluous. “Current disordered eating” refers to the participants’ (mothers’ and fathers’) disordered eating at the time of the study.

  1. L 22-23- Was it not expected/ anticipated? This is common sense. Suggest to start the sentence as—However, as expected, our findings reassure that parental CFP entrenches in our parental life and can therefore be passed on to our next generation.

We have changed the word “show” to “confirm”, which implies that this was known / expected previously.

  1. L29- Remove the first sentence of the Intro.

The first sentence of the Introduction has been removed.

  1. L 111-116- Again, how about male parents? Are you indicating breastfeeding? Make it CLEAR, right from the beginning (title).

We have added the number of male parents (fathers). We have changed the wording to emphasize that the children were aged between 2 and 18. Since it is extremely rare for children above the age of 2 to be breastfed, we believe it is clear that we are not referring to breastfeeding. However, if you believe that it is unclear, this information can be added: “at least one child between the age of 2 and 18 living with them at home, who is not breastfeeding”.

  1. L231- Fig 1. How did you prepare this? Is it your own? Or you followed the technique/ method from others and then inserted/applied your data/information to see what it says? If it is from others, please refer and cite them.

We ourselves prepared this Figure, which presents our data.

  1. L259-260- Fig 2. Add both axes to the Fig. Revise the name of X-axis. Parental disordered eating itself should be enough to understand that it would be high as the value of X-axis increases.

This has been corrected.

  1. L273- Fig 3. Same comments as given for Fig 2.

This has been corrected.

  1. L345- what does maternal stand for? Why? Clarify.

“Maternal” means used by one’s mother. We have now used the words “used by one’s mother”. We hope that our manuscript now makes it clearer that retrospectively recalled child feeding practices refer to participants’ childhood memories of the child feeding practices used by their mothers.

  1. L 369- Remove the first two words of Conclusions

The first two words of the Conclusions section have been removed.

  1. L376- Remove/ replace Sadly. This word is inappropriate for a study like yours.

The word “sadly” has been removed.

  1. *** Suggest to add/ provide all data as supplementary.

Editor: Please advise me as to how to include data.

Reviewer 2 Report

The manuscript presents a study of high interest and relevance, which assesses the association between child feeding practices between generations and their impact on the body mass index.
The study was well conducted and the results are conveniently presented and discussed.
The results of the study are of great importance to guide clinical and educational counseling in terms of child feeding practices for parents, showing that assessing adults' retrospective perception of their child feeding practices during their childhood can provide relevant data.

I present some minor revisions below.

l. 41-42: sentence could be better contextualized, perhaps incorporated in the sentence in lines 34-35.

Methods:

-    It could be important for the authors to describe in more detail the process of recruiting and selecting participants. It is said that it was through social media, but it could be more detailed.
-    It could be important for the authors to detail what sociodemographic characterization data were collected

Discussion

-    Authors stated that "BMI was not measured or reported because of the focus of our study was on parental perception of their children's weight" and considered it a limitation. However, it could be important in the discussion to develop this point further, albeit in hypothetical terms, bearing in mind that the real BMI could actually be a major confounder. 

Author Response

Review 2

The manuscript presents a study of high interest and relevance, which assesses the association between child feeding practices between generations and their impact on the body mass index.
The study was well conducted and the results are conveniently presented and discussed.
The results of the study are of great importance to guide clinical and educational counseling in terms of child feeding practices for parents, showing that assessing adults' retrospective perception of their child feeding practices during their childhood can provide relevant data.

I present some minor revisions below.

  1. 41-42: sentence could be better contextualized, perhaps incorporated in the sentence in lines 34-35.

We agree with you, and in re-reading lines 34-35 (in the previous manuscript), we have reached the conclusion that the last sentence of this paragraph can be omitted since the point has already been made.

Methods:

  1. It could be important for the authors to describe in more detail the process of recruiting and selecting participants. It is said that it was through social media, but it could be more detailed.

We have added the following: “…mostly via Facebook pages relevant to parenting and nutrition, intended for young, normative adults in a representative selection of geographical areas in Israel.”

  1. It could be important for the authors to detail what sociodemographic characterization data were collected

Unfortunately, the only sociodemographic variable that we collected was education. We have expanded on what we previously wrote about this.

Discussion

  1. Authors stated that "BMI was not measured or reported because of the focus of our study was on parental perception of their children's weight" and considered it a limitation. However, it could be important in the discussion to develop this point further, albeit in hypothetical terms, bearing in mind that the real BMI could actually be a major confounder. 

It is true that child’s BMI can be a consideration in parental feeding; however, perceived child weight is even more important for parental feeding practices which are the focus of the current paper. This is now emphasized in the discussion.

Reviewer 3 Report

Dear authors,

Thank you for giving me the opportunity to read and review this manuscript. I think that this is a very interesting article, I am sure that your paper will provide a relevant study about the topic. However, I suggest the following ammendments, in order to be finally accepted:

- It could be better to talk about “parents” not only the “mother” in all the document (also the title).

- Maybe the abstract could be rewritten in a more summary way (at the moment 240 words, but the maximum should be about 200).

-   There are some formal mistakes in the text. E.g. a) Line 47: “Pre-ocuppation”; b) Line 135: “Retrospective Child Feeding Practices were assessed by The CFQ was adapted to […]” it seems to be missing text between “by” and “The”; c) the template used is not the most updated (it appears year 2019); or d) Line 319: “[…] weight and eating attitudes”, there is a symbol before the word “eating”. It would be be advisable to carry out an in-depth review of the formal aspects of the document.

- Although the paper cites 39 references, It could be necessary to go a little deeper into the analysis of the state of the art, especially with the incorporation of more recent scientific studies (scientific articles or scientific books) on the subject (at least, about the theoretical general backgrounds).

Author Response

Review 3

Thank you for giving me the opportunity to read and review this manuscript. I think that this is a very interesting article, I am sure that your paper will provide a relevant study about the topic. However, I suggest the following amendments, in order to be finally accepted:

  1. It could be better to talk about “parents” not only the “mother” in all the document (also the title).

We agree and would have liked to present data about paternal child feeding practices in addition to maternal child feeding practices. However, we asked participants to report on their childhood recollections of the child feeding practices used by their mothers only, since mothers are, in general, more involved than fathers with feeding their children. We have tried to clarify this throughout the manuscript and have also noted it as a limitation of the study.

  1. Maybe the abstract could be rewritten in a more summary way (at the moment 240 words, but the maximum should be about 200).

The abstract has been shorted to under 200 words.

There are some formal mistakes in the text. E.g.

  1. Line 47: “Pre-ocuppation”

The hyphen has been deleted.

  1. Line 135: “Retrospective Child Feeding Practices were assessed by The CFQ was adapted to […]” it seems to be missing text between “by” and “The”

Thank you for picking up on this error. The sentence has been corrected.

  1. the template used is not the most updated (it appears year 2019); or

We are afraid that we have been unsuccessful in downloading the updated template, and would appreciate receiving assistance with this from the Editor.

  1. Line 319: “[…] weight and eating attitudes”, there is a symbol before the word “eating”. It would be advisable to carry out an in-depth review of the formal aspects of the document.

Again, thank you for catching this. We have deleted the symbol and have done our best to eliminate other technical issues throughout the manuscript.

  1. Although the paper cites 39 references, It could be necessary to go a little deeper into the analysis of the state of the art, especially with the incorporation of more recent scientific studies (scientific articles or scientific books) on the subject (at least, about the theoretical general backgrounds).

We have now mentioned two recent, additional studies in the context of theoretical background, in the second paragraph of the Introduction. The references are provided in comments in the manuscript. We did not use automated software to format the references, so that renumbering the references will take us considerable time. If the Editor would like us to renumber the references, we will invest the necessary time and effort.

Reviewer 4 Report

My comments are as follows:

  1. On the author list in a place of asterisk number 4 is added – for corresponding author
  2. The Likert scale is constructed as “-“, “0”, “+” so the neutral answer is in the center. If the answer value increasing from “0” to “5’” it is simple 6 points scale, but not Likert.
  3. In the line 119 the square bracket should be used for literature position with the link to the questionnaire used; line 157 as well.
  4. The authors presented at the end of the article a few limitations, but from my point of view it is more of them. 
  5. I have methodological doubts – have authors selected one person (mother or father)  from one family, or have both parents been examined? It is not explained but it plays an important role in the results of the questionnaire analizes.

Author Response

Review 4

  1. On the author list in a place of asterisk number 4 is added – for corresponding author.

This has been corrected.

  1. The Likert scale is constructed as “-“, “0”, “+” so the neutral answer is in the center. If the answer value increasing from “0” to “5’” it is simple 6 points scale, but not Likert.

This has been changed.

  1. In the line 119 the square bracket should be used for literature position with the link to the questionnaire used; line 157 as well.

This has been changed.

  1. The authors presented at the end of the article a few limitations, but from my point of view it is more of them. 

Two additional limitations have been added.

  1. I have methodological doubts – have authors selected one person (mother or father) from one family, or have both parents been examined? It is not explained but it plays an important role in the results of the questionnaire analyzes.

As now stated in the Participants subsection of the Methods, each family was represented by either the mother or the father (not both).
